# Increasing the Coverage and Balance of Robustness Benchmarks by Using Non-Overlapping Corruptions

## Abstract

Neural Networks are sensitive to various corruptions that usually occur in real-world applications such as blurs, noises, low-lighting conditions, etc. To estimate the robustness of neural networks to these common corruptions, we generally use a group of modeled corruptions gathered into a benchmark. We argue that corruption benchmarks often have a poor coverage: being robust to them only imply being robust to a narrow range of corruptions. They are also often unbalanced: they give too much importance to some corruptions compared to others. In this paper, we propose to build corruption benchmarks with only non-overlapping corruptions, to improve their coverage and their balance. Two corruptions overlap when the robustnesses of neural networks to these corruptions are correlated. We propose the first metric to measure the overlapping between two corruptions. We provide an algorithm that uses this metric to build benchmarks of Non-Overlapping Corruptions. Using this algorithm, we build from ImageNet a new corruption benchmark called ImageNet-NOC. We show that ImageNet-NOC is balanced and covers several kinds of corruptions that are not covered by ImageNet-C.

## 1 Introduction

Neural Networks perform poorly when they deal with images that are drawn from a different distribution than their training samples. Indeed, neural networks are sensitive to adversarial examples (Szegedy et al., 2014), background changes (Xiao et al., 2020), and common corruptions (Hendrycks & Dietterich, 2019).

Common corruptions are perturbations that change the appearance of images without changing their semantic content. For instance, neural networks are sensitive to noises (Koziarski & Cyganek, 2017), blurs (Vasiljevic et al., 2016) or lighting condition variations (Temel et al., 2017). Contrary to adversarial examples (Szegedy et al., 2014), common corruptions are not artificial perturbations especially crafted to fool neural networks. They naturally appear in industrial applications without any human interfering, and can significantly reduce the performances of neural networks.

A neural network is robust to a corruption $c$, when its performances on samples corrupted with $c$ are close to its performances on clean samples. Some methods have been recently proposed to make neural networks more robust to common corruptions (Geirhos et al., 2019; Hendrycks* et al., 2020; Rusak et al., 2020).

To determine whether these approaches are effective, it is required to have a method to measure the neural network robustness to common corruptions. The most commonly used method consists in evaluating the performances of neural networks on images distorted by various kinds of common corruptions: (Hendrycks & Dietterich, 2019; Karahan et al., 2016; Geirhos et al., 2019; Temel et al., 2017). In this study, we call the group of perturbations used to make the robustness estimation a corruption benchmark. We also use this term to refer to a set of test images that have been corrupted with these various corruptions. We identify two important factors that should be taken into account when building a corruption benchmark: the **balance** and the **coverage**.

In this paper, we consider that a corruption $c$ is covered by a benchmark, when increasing the robustness of a network to all the corruptions of this benchmark, also increases the robustness of

the network to $c$. For instance, a benchmark that contains a camera shake blur corruption covers the defocus blur corruption, because the robustnesses towards these two corruptions are correlated (Vasiljevic et al., 2016). The coverage of a benchmark is defined as the number of corruptions covered by this benchmark. The more a benchmark covers a wide range of common corruptions, the more it gives a complete view of the robustness of a neural network.

At the same time, we consider a benchmark as balanced when it gives the same importance to the robustness to every corruption it contains. For instance, according to a balanced benchmark, being robust to noises is as important as being robust to brightness variations. We argue that most of the existing corruption benchmarks are unbalanced: they give too much importance to the robustness to some corruptions compared to others.

The coverage and balance of corruption benchmarks are related to the notion of **corruption overlappings**. We say that two corruptions overlap when the robustnesses of neural networks towards these corruptions are correlated. The contribution of this paper is fourfold:

1. We propose the first method to estimate to what extent two corruptions overlap.
2. We show that building corruption benchmarks with non-overlapping corruptions make them more balanced and able to cover a wider range of corruptions.
3. We propose a method to build benchmarks that contain only non-overlapping corruptions.
4. We use this method to build from ImageNet, a benchmark of Non-Overlapping Corruptions called ImagNet-NOC, to estimate the robustness of image classifiers to common corruptions. We show that ImagNet-NOC is balanced and covers corruptions that are not covered by ImageNet-C: a reference corruption benchmark (Hendrycks & Dietterich, 2019).

## 2 BACKGROUND AND RELATED WORKS

### 2.1 ESTIMATING THE ROBUSTNESS OF NETWORKS WITH OUT-OF-DISTRIBUTION SAMPLES

Studying the performances of neural networks on samples that lie outside training distributions, is a widely studied domain. Being able to understand out-of-distribution (o.o.d) samples is essential to guarantee that neural networks are reliable in real-world applications. Several benchmarks and methods have been proposed to study this field. For instance, ImageNet-A (Dan Hendrycks & Song, 2019) is a simple benchmark for ImageNet classifiers that contains samples drawn from a different source than the one used to build ImageNet. Adversarial examples, are samples that have been slightly modified to fool neural networks (Szegedy et al., 2014). Making sure that models are robust to these kinds of o.o.d samples is essential in terms of security. Artistic renditions (Hendrycks et al., 2020) or sketches (Haohan et al., 2019), can also be useful to determine if neural networks understand the abstract concepts we want them to learn. Methods to study how classifiers are affected by background changes have also been recently proposed (Beery et al., 2018; Xiao et al., 2020).

Another important aspect of the robustness of neural networks to o.o.d samples, is the robustness to common corruptions. This aspect of the robustness is generally estimated by gathering several commonly encountered corruptions, and by testing the performances of neural networks on images corrupted with these corruptions. Diverse selections of common corruptions have been proposed to make a robustness estimation (Karahan et al., 2016; Laugros et al., 2019; Geirhos et al., 2019). In particular, ImageNet-C is a popular benchmark used to measure the robustness of ImageNet classifiers (Hendrycks & Dietterich, 2019). Different common corruption benchmarks have also been proposed in the context of object detection (Michaelis et al., 2019), scene classification (Tadros et al., 2019) or, eye-tracking (Che et al., 2020). It is worth noting that some transformations that are in between adversarial attacks and common corruptions have been recently proposed to measure the robustness of image classifiers (Kang et al., 2019; Dunn et al., 2019; Liu et al., 2019).

### 2.2 CORRUPTION OVERLAPPINGS IN BENCHMARKS

It has been noticed that fine-tuning a model with camera shake blur helps it to deal with defocus blur and conversely (Vasiljevic et al., 2016). The robustnesses to diverse kinds of noises have also been shown to be closely related (Laugros et al., 2019). Even for two corruptions that do not look similar to the human eye, increasing the robustness of a model to one of these corruptions, can

imply increasing the robustness to the other corruption (Kang et al., 2019). In general, it has been shown that the robustnesses to the corruptions that distort the high-frequency content of images are correlated (Yin et al., 2019). In the context of adversarial examples, it is known that the robustness towards one adversarial attack can be correlated with the robustness to another attack (Tramer & Boneh, 2019). So, it is generally recommended to evaluate the adversarial robustness with attacks that are clearly different from each other (Carlini et al., 2019). The experiments carried out in this paper suggest that this recommendation should also be followed in the context of common corruption robustness estimation.

## 3 Corruption Overlapping

### 3.1 The Corruption Overlapping Score

We consider that two corruptions overlap when the robustness to one of these corruptions is correlated with the robustness to the other corruption. In this section, we propose a methodology to estimate to what extent two corruptions overlap.

**The Robustness Score.** To determine whether two corruptions overlap, we first need to introduce a metric called the robustness score. This score gives an estimation of the robustness of a model $m$ to a corruption $c$. It is computed with the following formula: $R_c^m = \frac{A_c}{A_{clean}}$.

$A_{clean}$ is the accuracy of $m$ on an uncorrupted test set and $A_c$ is the accuracy of $m$ on the same test set corrupted with $c$. The higher $R_c^m$ is, the more robust $m$ is. Please note that using this metric requires to monitor $A_{clean}$ and make sure it is relatively high. Otherwise, an untrained model for which $A_c$ equals $A_{clean}$, would be considered as robust for example. In this study, this metric is used only in the methodology we propose to estimate the overlapping between two corruptions.

**The Corruption Overlapping Score.** We consider two neural networks $m1$ and $m2$ and two corruptions $c1$ and $c2$. $m1$ and $m2$ are identical, and trained with exactly the same settings except that their training sets are respectively augmented with the corruptions $c1$ and $c2$. A standard model is trained the same way but only with non-corrupted samples. We propose a method to measure to what extent $c1$ and $c2$ overlap. The idea of the method is to see if a data augmentation with $c1$ makes a model more robust to $c2$ and conversely. To determine this, $m1$, $m2$, and a test set are used to compute the following expression:

$$(R_{c1}^{m2} - R_{c1}^{standard}) + (R_{c2}^{m1} - R_{c2}^{standard}) \tag{1}$$

The first term of (1) measures whether a model that fits exactly $c2$ is more robust to $c1$ than the standard model. Symmetrically, the second term measures whether a model that fits exactly $c1$ is more robust than the standard model to $c2$. The more making a model fit $c1$ implies being more robust to $c2$ and reciprocally, and the more we can suppose that the robustnesses to $c1$ and $c2$ are correlated in practice. In other words, the expression (1) gives an estimation of the overlapping between $c1$ and $c2$. To be more convenient, we would like to build a corruption overlapping score equal to 1 when $c1 = c2$, and equal to 0 when the robustnesses to $c1$ and $c2$ are not correlated at all. We propose a new expression that respects both conditions:

$$O_{c_1,c_2} = \max\{0, \frac{1}{2} * \left( \frac{R_{c2}^{m1} - R_{c2}^{standard}}{R_{c2}^{m2} - R_{c2}^{standard}} + \frac{R_{c1}^{m2} - R_{c1}^{standard}}{R_{c1}^{m1} - R_{c1}^{standard}} \right)\} \tag{2}$$

The expression (2) is a normalized version of (1). It measures the overlapping between two corruptions while respecting the conditions mentioned above. Indeed, if a data augmentation with $c1$ does not increase the robustness to $c2$ at all and conversely, then the ratios in (2) are null or negative, so the whole overlapping score is maximized to zero. In other words, when $c1$ and $c2$ do not overlap at all, the overlapping score is equal to 0. Besides, when $c1 = c2$, $R_{c2}^{m1} = R_{c2}^{m2}$ and $R_{c1}^{m2} = R_{c1}^{m1}$, so both ratios of (2) are equal to 1. Then, $O_{c_1,c_2} = 1$ when $c1$ and $c2$ completely overlap.

**How to compute an overlapping score.** To get the overlapping score between $c1$ and $c2$, we follow the method illustrated in Figure 1. This method has six steps, and requires to have a training set, a test set and three untrained models that share the same architecture ($m1$, $m2$ and $standard$). The

step (1), consists in using the corruptions $c1$ and $c2$ to get two training sets, each corrupted with one corruption. Then, the obtained corrupted sets are used to train the models $m1$ and $m2$ in step (2). The standard model is also trained during this step but only with non-corrupted samples. In step (3), similarly to step (1), we use $c1$ and $c2$ to get two corrupted versions of the test set. The accuracies of the three models on the three test sets are computed in step (4). The scores obtained are used in step (5), to get the robustness scores of each model for the corruptions $c1$ and $c2$. The results obtained are used to compute the overlapping score between $c1$ and $c2$ in step (6).

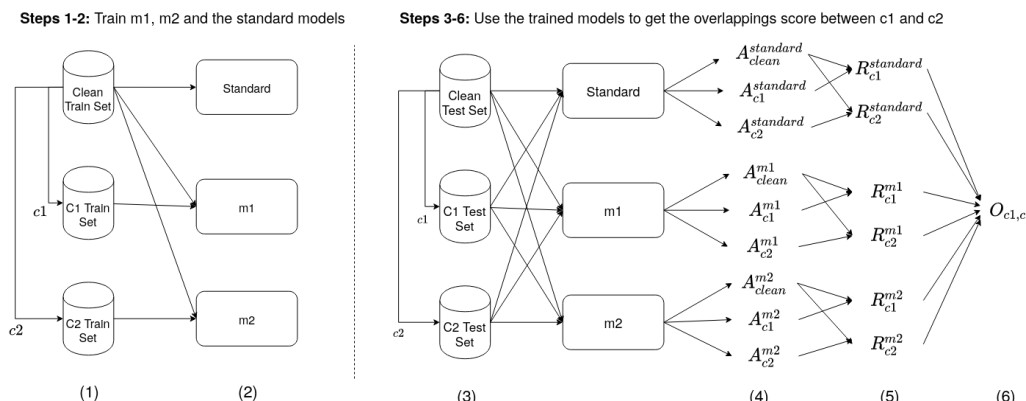

Figure 1: Methodology used to compute the overlapping score between two corruptions $c_1$ and $c_2$.

## 3.2 CORRUPTION OVERLAPPING AND COVERAGE OF BENCHMARKS

With our definition, a corruption $c$ is covered by a benchmark, when increasing the robustness of a network to all the corruptions of this benchmark, also increases the robustness of the network to $c$. The more a benchmark covers a wide range of corruptions, the more being robust to this benchmark provides a strong guarantee about the robustness of a neural network. Then, a benchmark should cover as much common corruptions as possible.

To illustrate the notion of coverage, let us consider *bench1*, a benchmark that contains three corruptions of ImageNet-C: Gaussian noise, shot noise and impulse noise (Hendrycks & Dietterich, 2019). We also consider *bench2*, that contains the Gaussian noise, brightness and elastic corruptions of ImageNet-C. Intuitively, being robust to *bench1* implies being robust only to noises while being robust to *bench2* implies being robust to a wider range of corruptions. Then, we can suppose that *bench1* has a lower coverage than *bench2*.

When we compute the overlapping scores of these benchmarks, we observe that the overlapping between the corruptions of *bench1* are close to 1, while they are close to 0 in *bench2* (see Figure 3). The corruptions of *bench1* clearly overlap while the ones of *bench2* do not. We argue that the overlappings in benchmarks tend to reduce their coverage. Indeed, the more two corruptions $c_1$ and $c_2$ overlap, the more it is likely that a corruption covered by $c_1$ is also covered by $c_2$ and conversely. So, when two corruptions overlap, their range of covered corruptions overlap too. By reducing the overlappings in a benchmark, we separate the ranges of corruptions covered by the corruptions of this benchmark, which results in increasing its coverage. In Section 5.2, we show that we can cover the fifteen corruptions of ImageNet-C with only eight non-overlapping corruptions.

## 3.3 CORRUPTION OVERLAPPING AND BALANCE OF BENCHMARKS

We consider that a benchmark is balanced, when it gives the same importance to the robustness to every corruption it contains. For instance in Section 5.3, we show that the ImageNet-C benchmark gives more importance to the blur corruptions than to the corruptions that affect the brightness of images. Yet, in a real-world applications, we think that being robust to different kinds of blurs is not more valuable than being robust to lighting condition variations. Being unbalanced is in general not a desirable property, it makes benchmarks give biased estimations of neural network robustness.

Overlappings between the corruptions of a benchmark can make it unbalanced. Let us consider three corruptions $c_1$, $c_2$, and $c_3$ with $c_1$ and $c_2$ that completely overlap, and $c_1$ and $c_2$ that do not overlap at all with $c_3$. A model robust to $c_3$, is robust to one third of the corruptions of the benchmark. But a model robust to $c_1$ is also robust to $c_2$, because $c_1$ and $c_2$ overlap. So being robust to $c_1$ or $c_2$ implies being robust to two third of the corruptions of the benchmark. Then, this benchmark rewards more the robustness to $c_1$ or $c_2$ than the robustness to $c_3$: it is unbalanced. In general, if one corruption contributes more to the total overlapping of a benchmark than another corruption of this benchmark, then the benchmark is unbalanced. In Section 5.3, we show that a benchmark built with non-overlapping corruptions is more balanced than ImageNet-C.

## 4 CONSTRUCTION OF A NON-OVERLAPPING CORRUPTION BENCHMARKS: IMAGENET-NOC

**Experimental Set-up.** For every training of this study, we use the following parameters. The used optimizer is SGD with a momentum of 0.9. The used cost function is a cross-entropy function, with a weight decay set to $10^{-4}$. Models are trained for 40 epochs with a batch size of 256. The initial learning rate is set to 0.1 and is divided by 10 at epoch 20 and 30. In all the experiments we use ImageNet-100: a subset of ImageNet that contains every tenth ImageNet class by WordNetID order (Deng et al., 2009). All images are resized to the 224x224 format, and randomly horizontally flipped with a probability of 0.5 during trainings. When we use a data augmentation with a corruption in a training, half of the images of each training batch are transformed with the corruption, while the other half is not corrupted.

We present Algorithm 1: a general method to build benchmarks that do not contain any overlapping corruption. We argue that this method helps to build balanced benchmarks that have a large coverage.

---

**Algorithm 1 Methodology Proposed to Build a Benchmark of Non-Overlapping Corruptions**

**Require:** $S$ a set of common corruptions
**Require:** A $train\ set$, a $test\ set$ and a neural network $architecture$
**Require:** An $overlapping\ threshold$: the maximum overlapping score allowed in the benchmark
  (0) $n \leftarrow 2$. $n$ is the current number of corruptions in the benchmark. It is initialized to 2.
  (1) Use the $train\ set$, the $test\ set$ and the chosen network $architecture$ to apply the methodology presented in Section 3.1, to get all the overlapping scores between the corruptions of S.
  (2) Pick the largest subsets of S with overlapping scores under the $overlapping\ threshold$.
  (3) Among the retained subsets, select the one with the lowest mean overlapping score to form the benchmark.

---

We want to use this algorithm to build a new benchmark that measures the robustness of image classifiers to common corruptions. Algorithm 1 requires to gather a group of candidate corruptions called $S$. The larger $S$, the more combinations of non-overlapping corruptions can be found in $S$, the larger the benchmarks built by the algorithm. Then, we recommend to use a large intial set of corruptions to increase the coverage of the built benchmarks. For this study, we implemented two dozens of image corruptions (illustrated in Figure 2) to constitute $S$. All these corruptions are associated with a severity range. A value is randomly chosen from the severity range of the considered corruption each time an image is corrupted. The higher this value is, the more the aspect of the corrupted image changes. More information about the modeled common corruptions can be found in Appendix A.

We apply Algorithm 1, using this set of corruptions, the ImageNet $train\ set$ and $test\ set$, and the ResNet-18 $architecture$; with different values of the $overlapping\ threshold$. The corruption benchmarks obtained for different values of the threshold are shown in Appendix B. The higher the overlapping threshold is, the more the number of corruptions included in the constructed benchmarks increases, and so does the coverage of the constructed benchmarks. However, the coverage gain due to the increase of the threshold is reduced by overlappings, because overlapping corruptions tend to cover the same kind of corruptions. Besides, as explained in Section 3.3, the more there are overlappings in a benchmark, the more it is likely to be unbalanced. All in all, selecting the overlapping threshold, determines when the coverage gain does not worth the balance loss. Choosing this value depends on the application case and the kind of robustness estimation we want to make.

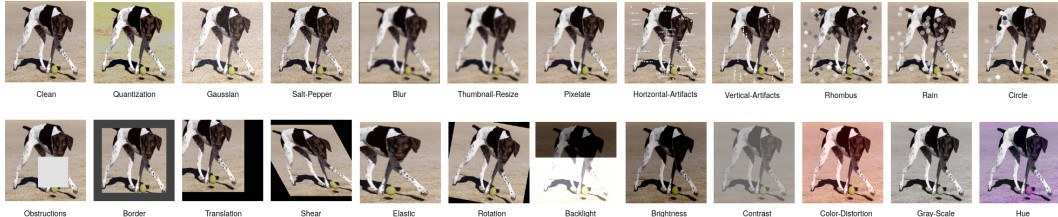

Figure 2: Illustrations of the common corruptions gathered to run the Algorithm 1.

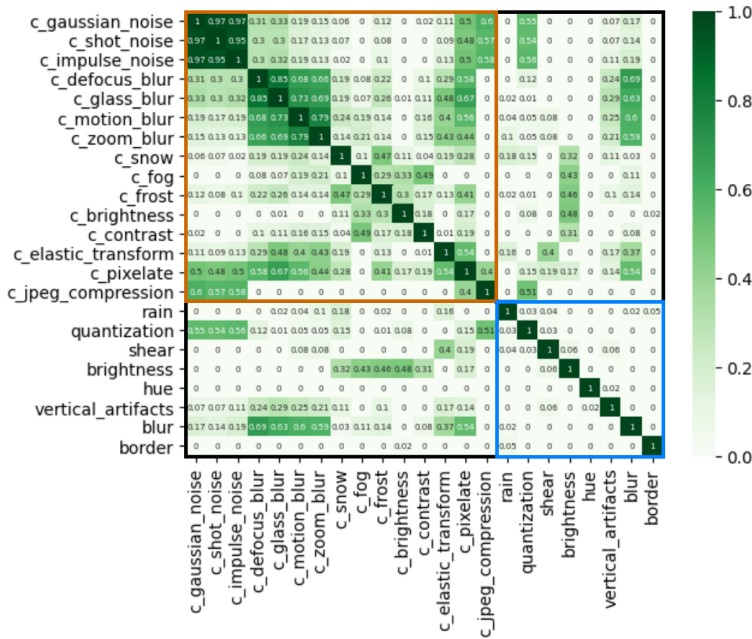

Figure 3: The overlapping scores between all the ImageNet-NOC and ImageNet-C corruptions.

Each benchmark obtained with Algorithm 1 that contains $n$ corruptions, are the $n$-corruption benchmarks with the lowest mean overlapping as possible. As explained in Sections 3.2 and 3.3, we expect that benchmarks that are optimal in terms of overlapping will have a good balance and coverage. We propose to study the set of eight corruptions obtained with $overlapping\ threshold = 0.1$, which is: *rain, quantization, shear, brightness, hue, vertical_artifacts, blur* and *border*. The overlapping scores between these corruptions are displayed in the right lower square in Figure 3. Eight corrupted ImageNet validation sets, each corrupted with one of the eight corruptions, are gathered to form **ImageNet-NOC**.

We run the algorithm a second time using a DenseNet-121 architecture instead of the ResNet-18 one. The overlapping scores obtained by running the step (1) of Algorithm 1 are displayed in Appendix A. The benchmark obtained with $overlapping\ threshold = 0.1$ is *rain, Gaussian, shear, brightness, hue, vertical_artifacts, elastic*. This benchmark shares six corruptions with ImageNet-NOC, and the overlapping score computed with *Gaussian noise* and *quantization* equals 0.6 and the one computed with *elastic* and *blur* equals 0.26. So, the only two corruptions that are not shared by the two benchmarks appear to be correlated in terms of robustness. So, using a DenseNet-121 architecture makes the algorithm build a benchmark that is fairly similar to the one obtained using ResNet-18.

Running the Algorithm 1 requires to complete one training for each corruption in $S$. It took one week with a single GPU Nvidia Tesla V100 to get all the overlapping scores of Figure 3. While this computational cost is high, we think that the process could be accelerated by fine-tuning models for a few epochs instead of training them from scratch. Further investigations should be conducted to determine to what extent this alternative would modify the obtained results.

**How to Use ImageNet-NOC**. We recommend to use the CE metric (Hendrycks & Dietterich, 2019) to measure the robustness of an image classifier to an ImageNet-NOC corruption. The mean CE score computed with a benchmark is called mCE. Using the mCE metric avoids several pitfalls while measuring the robustness of neural networks. To compute a CE score, it is required to get the error rate of a pretrained AlexNet model, on the ImageNet validation set corrupted with the considered corruption. The error rates of the torchvision pretrained AlexNet computed with the corruptions introduced in this paper are displayed in Appendix A. We provide the ImageNet-NOC CE scores of some traditionally used ImageNet classifiers in Appendix C. More details about how to use ImageNet-NOC can be found by visiting [Link available upon acceptance].

## 5 Comparison between ImageNet-NOC and ImageNet-C

### 5.1 Corruption Overlappings in ImageNet-C and ImageNet-NOC

ImageNet-C is a benchmark commonly used to measure the robustness of ImageNet classifiers to common corruptions (Hendrycks & Dietterich, 2019). It is built on fifteen common corruptions called Gaussian noise, shot noise, impulse noise, defocus blur, glass blur, motion blur, zoom blur, snow, frost, fog, brightness, contrast, elastic, pixelate, and jpeg compression. Each corruption is associated with five severity levels that determine to what extend the corrupted images are distorted. Please note that in general, the benchmark corruptions should not be used during a training of a model. Indeed, corruption benchmarks are built to estimate the robustness to unforeseen corruptions. In this study, we use the ImageNet-C and ImageNet-NOC corruptions during some training phases only because we analyze the benchmarks themselves.

Using the ResNet-18 model, the ImageNet training and validation sets, we apply the method illustrated in Figure 1 to get the overlapping score of every couple of ImageNet-C corruptions. The corruption severity is randomly selected for each image corrupted in the process. The obtained scores are displayed in the upper left square of Figure 3. We observe that all the corruptions that damage the textures in images (blurs, noises, pixelate and jpeg compression) significantly overlap. This result is consistent with the experiments carried out by Yin et al. (2019): they argue that the robustnesses of neural networks to corruptions that alter high-frequency information of images are correlated. Concerning the corruptions that alter low-frequency information of images, the overlappings are less pronounced. But we do observe some significant overlappings. There is a clear overlapping between fog and contrast or between snow and frost. As explained in Section 3.2 and 3.3, all these overlappings suggest that ImageNet-C is unbalanced and has a poor coverage. Figure 3 reveals that ImageNet-NOC contains far less overalpping corruptions than ImageNet-C.

We compute again the overlapping scores of ImageNet-C and ImageNet-NOC with the DenseNet-121 (Huang et al., 2017) and WideResNet-50-2 (Zagoruyko & Komodakis, 2016) architectures. The aspect of the overlapping arrays obtained with these two architectures is the same as the one obtained with ResNet-18 (see Appendix D). For traditionally used image classifiers, it appears that overlapping scores do not vary much with the architecture of the model used to compute them.

### 5.2 Coverage of ImageNet-NOC and ImageNet-C

In the lower left square of Figure 3, are displayed all the overlapping scores computed with one ImageNet-C corruption and one ImageNet-NOC corruption. We observe that for every ImageNet-C corruption $c1$, there is always at least one ImageNet-NOC corruption $c2$, for which the overlapping score computed with $c1$ and $c2$ is higher than 0.3. On the other hand, two ImageNet-NOC corruptions (*hue* and *border*) do not overlap at all with any of the ImageNet-C corruptions. Then, increasing the robustness to all the ImageNet-NOC corruptions should imply being more robust to all the ImageNet-C corruptions, but being robust to ImageNet-C may not imply being robust to some ImageNet-NOC corruptions.

To confirm this, we train two ResNet-18 called $m_{INOC}$ and $m_{IC}$. A data augmentation procedure with all the ImageNet-C corruptions is used to train $m_{IC}$. Each corrupted image of this training is modified by one randomly selected corruption of ImageNet-C, with a randomly chosen severity. Similarly, $m_{INOC}$ is trained with a data augmentation procedure with all the ImageNet-NOC corruptions. After the trainings, we measure the robustness of $m_{INOC}$ to every corruption of ImageNet-C by computing its CE scores. We also measure the CE scores of $m_{IC}$ towards the ImageNet-NOC

Table 1: Upper Table: the CE scores of $m_{INOC}$ on ImageNet-C (the lower is the better).
Lower Table: the CE scores of $m_{IC}$ on the ImageNet-NOC corruptions.

| | Clean | Gauss | Shot | Impul | Defo | Glass | Motion | Zoom | Snow | Frost | Fog | Bright | Contr | Elastic | Pixel | Jpeg | mCE |
|---|---|---|---|---|---|---|---|---|---|---|---|---|---|---|---|---|---|
| standard | 0.247 | 120 | 119 | 121 | 147 | 121 | 117 | 107 | 97 | 108 | 97 | 93 | 115 | 67 | 78 | 56 | 104 |
| $m_{INOC}$ | 0.199 | 92 | 94 | 93 | 102 | 112 | 86 | 78 | 69 | 81 | 67 | 41 | 97 | 63 | 71 | 53 | 81 |

| | Clean | Quant | Blur | Vert-Arti | Rain | **Border** | Shear | Bright | **Hue** | mCE |
|---|---|---|---|---|---|---|---|---|---|---|
| standard | 0.247 | 87 | 122 | 77 | 71 | **67** | 79 | 117 | **88** | 89 |
| $m_{IC}$ | 0.239 | 44 | 15 | 39 | 58 | **79** | 49 | 72 | **117** | 59 |

(2.a) The ImageNet-C mCE scores of models, each trained with one corruption of ImageNet-C.

| | Gauss | Shot | Impul | Defo | Glass | Motion | Zoom | Snow | Frost | Fog | Bright | Contr | Elastic | Pixel | Jpeg |
|---|---|---|---|---|---|---|---|---|---|---|---|---|---|---|---|
| mCE | 71 | 71 | 71 | 54 | 56 | 63 | 68 | 83 | 79 | 86 | 89 | 78 | 92 | 85 | 94 |

(2.b) The ImageNet-NOC mCE scores of models, each trained with one corruption of ImageNet-NOC.

| | Quant | Blur | Vert-Arti | Rain | Border | Shear | Bright | Hue |
|---|---|---|---|---|---|---|---|---|
| mCE | 83 | 79 | 83 | 83 | 81 | 90 | 79 | 88 |

corruptions. We compare the obtained scores of both models with the ones of the standard model in Table 1. The first column of this table contains the error rates on non-corrupted ImageNet samples.

We observe in Table 1 that $m_{INOC}$ is more robust than the standard model to all the ImageNet-C corruptions. However, $m_{IC}$ is not robust to the *hue* and *border* corruptions of ImageNet-NOC. These results appear to confirm the hypothesis made by studying the overlapping scores in Figure 3: ImageNet-C does not cover some of the ImageNet-NOC corruptions while ImageNet-NOC covers the ImageNet-C corruptions. As argued in Section 3.2, it seems that using non-overlapping corruptions helps to build benchmarks that have a larger coverage.

## 5.3 Balance of ImageNet-NOC and ImageNet-C

We carry out an experiment to compare the balance of ImageNet-NOC and ImageNet-C. We train one ResNet-18 for each corruption of ImageNet-NOC and ImageNet-C. So, fifteen models are trained using a data augmentation procedure with one corruption of ImageNet-C, and eight others are trained with one corruption of ImageNet-NOC. Then, we estimate the robustness of the first fifteen ResNet-18 on ImageNet-C by computing their mCE scores. We also get the mCE scores of the eight remaining ResNet-18 on ImageNet-NOC. The obtained scores are displayed in Tables 2.a, 2.b. The CE scores computed to get the mCE scores can be found in Appendix E.

The mCE scores obtained in Table 2.a are very different from each other. For instance, the mCE score of the model trained with *defocus_blur*, is much lower than the one trained with *brightness*. Then, according to the robustness estimation made with ImageNet-C, one of the models is much more robust than the other one. In other words, the ImageNet-C benchmark gives more importance to the robustness to *defocus_blur* than to the robustness to *brightness*: ImageNet-C is unbalanced.

We observe far less variations in the mCE scores in Table 2.b than in Table 2.a. More precisely, the difference between the lowest mCE and the highest mCE in Table 2.a and 2.b are respectively 40 and 11. And the standard deviation of the mCE scores in these tables are respectively 12.1 and 3.7. Then, the importance given by ImageNet-C to the robustness of its corruptions, varies a lot with the considered corruption. This variation is way less important for our benchmark. Then, ImageNet-NOC is significantly more balanced than ImageNet-C.

## 5.4 Robustness Estimations Using ImageNet-NOC and ImageNet-C

The experiments carried out in Section 5.2 and 5.3, suggest that ImageNet-NOC has a better balance and coverage than ImageNet-C. To determine whether using ImageNet-NOC instead of ImageNet-C makes a difference in practice, we propose to compare the performances of various models on the two benchmarks. First, we measure the mCE scores of several pretrained torchvision classifiers

on both ImageNet-C and ImageNet-NOC. The results are displayed in Table 3 and the details of the computed CE scores can be found in Appendix C. We observe that some models are considered relatively robust to ImageNet-NOC but not robust to ImageNet-C. For instance, the VGG-19, ResNet-50, and WideResNet-50-2 are much more robust to ImageNet-NOC than the AlexNet model, but they are less robust than AlexNet to ImageNet-C.

Table 3: ImageNet-C (IC) and ImageNet-NOC (INOC) mCE obtained with several pretrained classifiers.

|  | Alex | Squeeze | VGG-11 | VGG-19-BN | Res-18 | Res-50 | Dense-121 | Dense-201 | Wide-Res-50 |
|---|---|---|---|---|---|---|---|---|---|
| mCE_IC | 100 | 118 | 123 | 111 | 104 | 105 | 94 | 89 | 101 |
| mCE_INOC | 100 | 106 | 110 | 88 | 86 | 81 | 73 | 66 | 78 |

The second experiment is carried out using four ResNet-50 that have been shown to be robust to ImageNet-C. These models are called *SIN+IN* (Geirhos et al., 2019), $ANT^{3x3}$ (Rusak et al., 2020), *Augmix* (Hendrycks* et al., 2020) and *DeepAugment* (Hendrycks et al., 2020). We measure the robustness of these models to ImageNet-NOC by computing their CE scores. Then, we compare these scores with the ones obtained with the torchvision pretrained ResNet-50 in Table 4. We observe that the *SIN+IN*, $ANT^{3x3}$, and *DeepAugment* models are not robust to *border*. Interestingly, we show in Section 5.2 that this corruption is not covered by ImageNet-C. This result confirms that ImageNet-NOC can reveal a low robustness of models to corruptions that are not covered by ImageNet-C.

The ImageNet-NOC and ImageNet-C mCE scores of the five considered models are compared in the two last columns of Table 4. We observe that the robustness ranking established with ImageNet-C is different from the one established with ImageNet-NOC. For instance, $ANT^{3x3}$ is very robust to ImageNet-C but not robust to ImageNet-NOC. We think this is a direct consequence of the lack of balance of ImageNet-C. Indeed, we provide evidence in Section 5.3 that ImageNet-C gives a lot of importance to the noise robustness, and $ANT^{3x3}$ has been shown to be particularly robust to noises (Rusak et al., 2020). So $ANT^{3x3}$ is considered as very robust to ImageNet-C, but not to ImageNet-NOC which is more balanced. We note that *Augmix* obtains a relatively low ImageNet-NOC mCE compared to the ImageNet-C one. This result should be considered cautiously because shears and quantizations are used by the Augmix data augmentation procedure, and these corruptions overlap with some of the ImageNet-NOC corruptions. This is a reason why the *Augmix* model obtains low *shear* and *quantization* CE scores.

The experiments carried out in this section show that the robustness estimations made with ImageNet-NOC are different from the ones made with ImageNet-C. We think that ImageNet-NOC should be preferred to ImageNet-C because of its coverage and balance.

# 6 CONCLUSION

We proposed a metric called the corruption overlapping score, that measures to what extend the robustnesses towards two corruptions are correlated. We showed that the overlappings between the corruptions of a benchmark can reduce its coverage and make it unbalanced. We provided a benchmark of Non-Overlapping Corruptions called ImageNet-NOC to measure the robustness of image classifiers. We showed that ImageNet-NOC is balanced and covers several kinds of common corruptions that are not covered by ImageNet-C. ImageNet-NOC is built thanks to the method we proposed to construct non-overlapping corruption benchmarks. This method can be easily adapted to other computer vision tasks. We hope it will be used to build other non-overlapping corruption benchmarks, that will help to make better estimations of the robustness of neural networks.

Table 4: ImageNet-NOC CE scores obtained for several data augmentation strategies. The two last columns contain the ImageNet-NOC and ImageNet-C mCE scores, and the associated ranks written in brackets.

|  | Quant | Blur | Vert-Arti | Rain | **Border** | Shear | Bright | Hue | INOC_mCE (rank) | IC_mCE (rank) |
|---|---|---|---|---|---|---|---|---|---|---|
| Standard | 130 | 60 | 95 | 93 | 53 | 87 | 71 | 58 | 81 (5) | 77 (5) |
| SIN+ | 85 | 55 | 74 | 86 | **56** | 82 | 61 | 49 | 68 (3) | 69 (4) |
| Augmix | 51 | 13 | 72 | 90 | 49 | 71 | 62 | 56 | 58 (1) | 65 (3) |
| $ANT^{3x3}$ | 85 | 56 | 80 | 109 | **58** | 76 | 67 | 55 | 73 (4) | 63 (2) |
| DeepAugment | 65 | 25 | 69 | 93 | **61** | 67 | 46 | 43 | 59 (2) | 60 (1) |

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

# A    PRESENTATION OF THE MODELED COMMON CORRUPTIONS

The common corruptions gathered in Section 4 are implemented with computationally cheap image transformations that can be easily added in an image pipeline. We provide in Figures 5 and 6, information about how the common corruptions used in this study are modeled. These corruptions are implemented for images that have pixel values in [0-1].

The *Severity Range* column of these arrays precises how the severity of each corruption is set. The lower bound of each severity range is selected to get a robustness score of 0.95 with the standard ResNet-18 model, tested on the ImageNet validation set that has been altered with the considered corruption. The upper bound is selected to get a robustness score of 0.5 in the same conditions. The upper bounds of the *hue* and *gray_scale* corruptions are different because these corruptions are not harmful enough to reach a robustness score of 0.5. They respectively reach a robustness score of 0.63 and 0.70 for the standard ResNet-18. The last column of the Figures 5 and 6 corresponds to the error rate of the torchvision pretrained AlexNet on the ImageNet validation set corrupted with the corruption indicated in the first column.

In Section 4, by completing the step (1) of Algorithm 1, we compute the overlapping scores between the corruptions gathered in Figure 2. These overlapping scores are displayed in Figure 4.

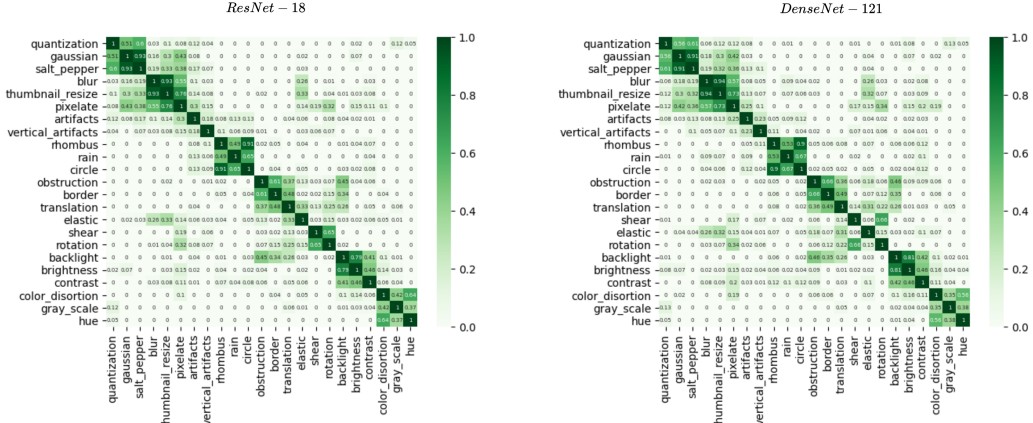

Figure 4: Overlapping scores between all the common corruptions displayed in Figure 2. The scores have been computed with the ResNet-18 and DenseNet-121 architectures

| Corruption Name | Illustration | Description | Severity Range | AlexNet Error Rate |
|---|---|---|---|---|
| Quantization | | Quantize the pixel values into a few quantization levels | Number of levels: 4 to 9 | 0.592 |
| Gaussian Noise | | Add zero mean Gaussian noise to the image | Standard deviation: 0.05 to 0.18 | 0.620 |
| Salt Pepper Noise | | Add salt-pepper noise in the image | Propability of each pixel to be changed: 0.003 to 0.032 | 0.649 |
| Blur | | Interpolate the image with its equivalent blured with five convolution with a 3x3 filter. The filter is filled is the value 1. | Interpolation factor associated to the blured image: 0.4 to 0.95 | 0.570 |
| Thumbnail Resize | | Apply a bilinear interpolation twice. Firstly to reduce the size of the image. Secondly to get back to its original size. | Reduction factor: 1.1 to 3.25 | 0.574 |
| Pixelate | | Convolve the image with a mean pooling filter | Size of the filter and padding used: 2 to 4 | 0.760 |
| Artifacts | | Add artifacts into the image. The artifacts are made with a small dotted lines | Number of artifacts: 15 to 170 | 0.746 |
| Vertical Artifacts | | The same corruption than Artifacts except that the artifacts are rotated by 90 degrees | Number of artifacts: 15 to 180 | 0.758 |
| Rhombus | | Several randomly selected 7x7 rhombus like areas are each filled with a unique pixel value | Number of rhombus: 9 to 76 | 0.728 |
| Rain | | Randomly select circle areas (7 radius circles). For these areas, add 1 to the pixel values and divide the result by 2 | Number of circles: 12 to 120 | 0.748 |
| Circles | | Several randomly selected circle areas (7 radius circles) are each filled with a unique pixel value | Number of circles: 7 to 50 | 0.720 |
| Obstruction | | One randomly chosen square area is filled with a unique pixel value | Size of the square 47 to 125 | 0.597 |

Figure 5: Presentation of half of the group of common corruptions displayed in Figure 2. The other half is presented in Figure 6.

| Corruption Name | Illustration | Description | Severity Range | AlexNet Error Rate |
|---|---|---|---|---|
| Border |  | Fill the border of images with a unique pixel value | Thickness of the corrupted area: 9 to 46 pixels | 0.670 |
| Translation |  | One horizontal and one vertical translation | Number of pixels translated for both translations: 15 to 62 | 0.603 |
| Shear |  | Horizontal Shear | Maximum pixel translation due to the shear: 0.39 | 0.657 |
| Elastic |  | Crop the height or the width of the image. The image is then resized to its original size | Number of cropped pixel: 44 to 110 | 0.541 |
| Rotation |  | Clockwise or anticlockwise rotation | Degree of the rotation: 7 to 50 | 0.711 |
| Backlight |  | Split the image into two areas and select a value. Add this value to all pixel of one of the area and substract the value for the other. | Value added or subtracted: 0.11 to 0.44 | 0.591 |
| Brightness |  | Select a unique value. Add this value or substract it to all the pixel value of the image | Value added or subtracted: 0.16 to 0.51 | 0.577 |
| Contrast |  | Reduce the contrast of images | Contrast reduction factor: 0.33 to 0.74 | 0.576 |
| Color Distortion |  | Add a unique value to all pixels of one of the RGB channel | Value added: 0.09 to 0.40 | 0.606 |
| Gray Scale |  | Interpolate the image with its gray scale version | Interpolation factor associated with the gray scale image: 0.49 to 1 | 0.580 |
| Hue |  | Shift cyclically the value of the pixels of the channel H in the HSV format | Value of the shifting: 0.05 to 0.5 | 0.684 |

Figure 6: Presentation of half of the group of common corruptions displayed in Figure 2. The other half is presented in Figure 5.

# B BENCHMARKS OBTAINED FOR DIFFERENT VALUES OF THE OVERLAPPING THRESHOLD

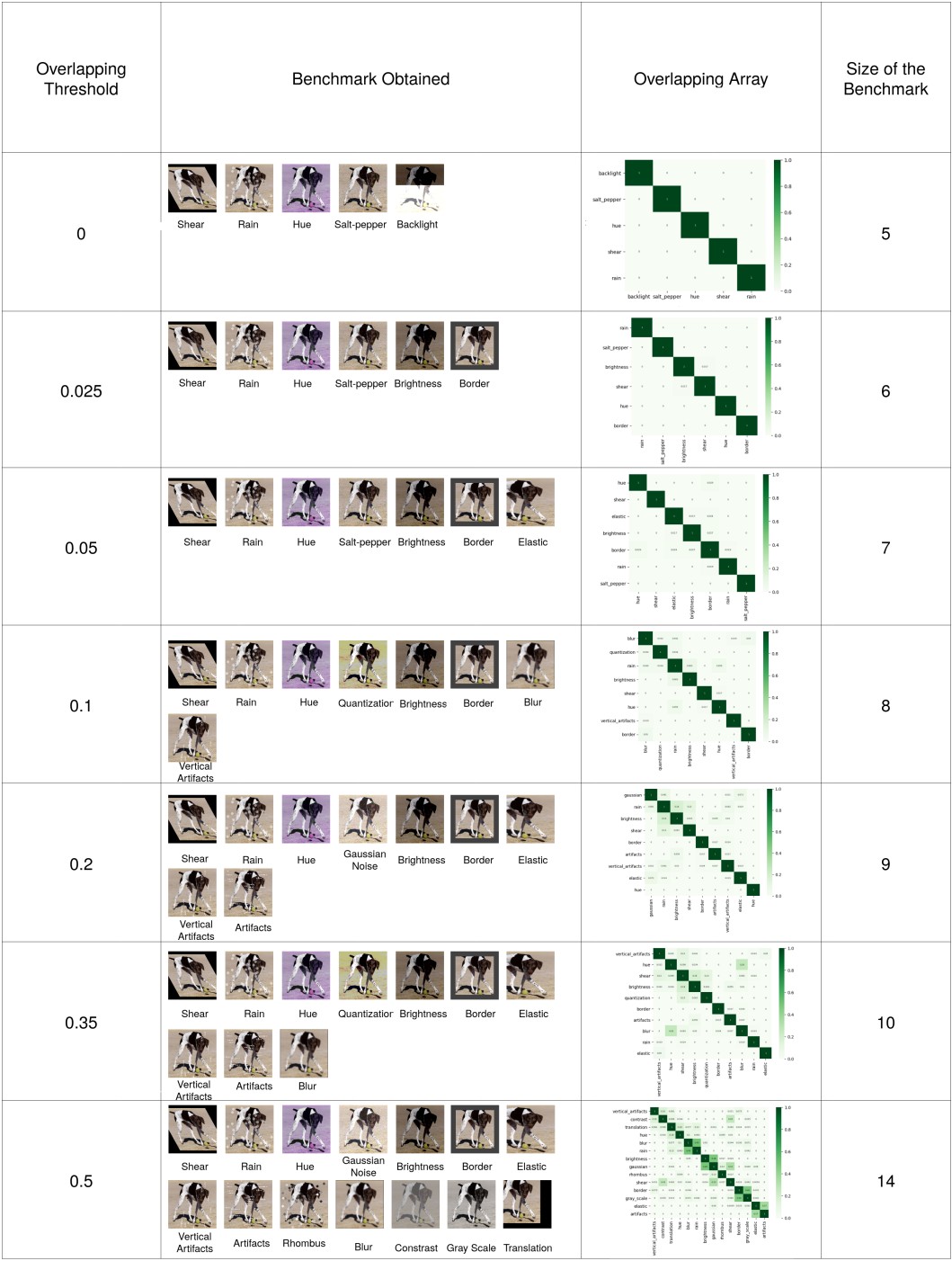

Figure 7: Benchmarks obtained when running Algorithm 1, for different values of the overlapping threshold. The third column contains the overlapping scores between the corruptions of the benchmarks.

## C  PERFORMANCES OF VARIOUS MODELS ON IMAGENET-NOC AND IMAGENET-C

Table 5: The ImageNet-C CE scores obtained with several pretrained torchvision classifiers.

|  | Gauss | Shot | Impul | Defo | Glass | Motion | Zoom | Snow | Frost | Fog | Bright | Contr | Elastic | Pixel | Jpeg | mCE |
|---|---|---|---|---|---|---|---|---|---|---|---|---|---|---|---|---|
| AlexNet | 100 | 100 | 100 | 100 | 100 | 100 | 100 | 100 | 100 | 100 | 100 | 100 | 100 | 100 | 100 | 100 |
| SqueezeNet | 118 | 116 | 114 | 104 | 110 | 106 | 105 | 106 | 110 | 98 | 101 | 100 | 126 | 129 | 229 | 118 |
| VGG-11 | 122 | 121 | 125 | 116 | 129 | 121 | 115 | 114 | 113 | 99 | 86 | 102 | 151 | 161 | 174 | 123 |
| VGG-19-BN | 104 | 105 | 114 | 108 | 132 | 114 | 119 | 102 | 100 | 79 | 68 | 89 | 165 | 125 | 144 | 111 |
| ResNet-18 | 104 | 106 | 111 | 100 | 116 | 108 | 112 | 103 | 101 | 89 | 67 | 87 | 133 | 97 | 126 | 104 |
| ResNet-50 | 104 | 107 | 107 | 97 | 126 | 107 | 110 | 101 | 97 | 79 | 62 | 89 | 146 | 111 | 132 | 105 |
| DenseNet-121 | 84 | 87 | 89 | 97 | 120 | 101 | 104 | 91 | 87 | 61 | 50 | 66 | 147 | 97 | 122 | 94 |
| DenseNet-201 | 82 | 87 | 87 | 91 | 116 | 100 | 107 | 85 | 82 | 63 | 45 | 62 | 136 | 80 | 113 | 89 |
| WideResNet-50-2 | 91 | 93 | 97 | 93 | 123 | 103 | 109 | 104 | 95 | 81 | 65 | 89 | 146 | 99 | 126 | 101 |

Table 6: The ImageNet-NOC CE scores obtained with several pretrained torchvision classifiers.

|  | Clean | Quant | Blur | Vert-Arti | Rain | Border | Shear | Bright | Hue | mCE |
|---|---|---|---|---|---|---|---|---|---|---|
| AlexNet | 43.5 | 100 | 100 | 100 | 100 | 100 | 100 | 100 | 100 | 100 |
| SqueezeNet | 41.8 | 143 | 97 | 109 | 111 | 93 | 99 | 109 | 86 | 106 |
| VGG-11 | 31.0 | 193 | 90 | 156 | 94 | 70 | 107 | 92 | 77 | 110 |
| VGG-19-BN | 25.8 | 175 | 70 | 113 | 77 | 55 | 91 | 71 | 53 | 88 |
| ResNet-18 | 30.2 | 128 | 68 | 103 | 84 | 69 | 92 | 80 | 64 | 86 |
| ResNet-50 | 23.9 | 130 | 60 | 95 | 93 | 53 | 87 | 71 | 58 | 81 |
| DenseNet-121 | 25.3 | 101 | 58 | 98 | 89 | 55 | 72 | 57 | 54 | 73 |
| DenseNet-201 | 22.8 | 89 | 54 | 86 | 84 | 49 | 69 | 52 | 50 | 66 |
| WideResNet-50-2 | 21.5 | 112 | 59 | 97 | 92 | 35 | 96 | 72 | 58 | 78 |

# D    OVERLAPPINGS OBTAINED FOR THREE DIFFERENT ARCHITECTURES

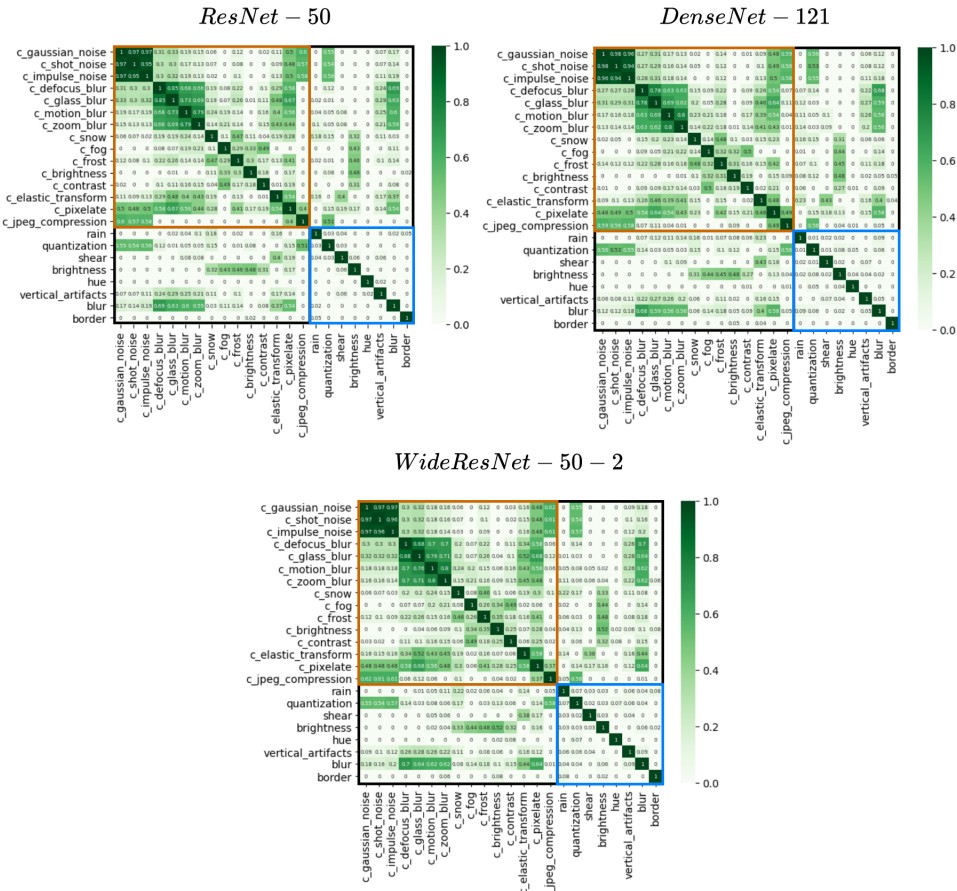

Figure 8: The overlapping scores between all the ImageNet-NOC and ImageNet-C corruptions, computed with the ResNet-50, DenseNet-121 and Wide-ResNet-50-2 architectures .

## E  PERFORMANCES OF MODELS TRAINED WITH A DATA AUGMENTATION WITH ONE CORRUPTION ON IMAGENET-C AND IMAGENET-NOC

We provide in Table 7 and 8, the CE scores computed and averaged to get the mCE scores of Tables 2.a and 2.b. The first column of these arrays contains the error rate on the non-corrupted ImageNet validation set.

Table 7: CE scores computed with models trained with a data augmentation with one corruption of ImageNet-C. Each line refers to one model trained with one corruption of ImageNet-C and each column refers to one corruption of ImageNet-C.

| | Clean | Gauss | Shot | Impul | Defo | Glass | Motion | Zoom | Snow | Frost | Fog | Bright | Contr | Elastic | Pixel | Jpeg | mCE |
|---|---|---|---|---|---|---|---|---|---|---|---|---|---|---|---|---|---|
| Standard | 0.247 | 121 | 119 | 122 | 147 | 121 | 118 | 107 | 97 | 109 | 98 | 94 | 116 | 68 | 79 | 56 | 104 |
| Gauss | 0.264 | 6 | 9 | 8 | 122 | 88 | 100 | 95 | 87 | 93 | 125 | 107 | 129 | 58 | 19 | 15 | 71 |
| Shot | 0.269 | 10 | 6 | 9 | 125 | 92 | 102 | 96 | 84 | 94 | 126 | 103 | 128 | 59 | 20 | 16 | 71 |
| Impul | 0.272 | 10 | 11 | 4 | 123 | 88 | 99 | 96 | 91 | 93 | 127 | 102 | 129 | 54 | 17 | 14 | 71 |
| Defo | 0.300 | 62 | 62 | 63 | 6 | 25 | 36 | 34 | 76 | 77 | 82 | 91 | 93 | 46 | 2 | 50 | 54 |
| Glass | 0.287 | 74 | 73 | 74 | 24 | 8 | 32 | 30 | 81 | 81 | 83 | 95 | 99 | 34 | 2 | 55 | 56 |
| Motion | 0.263 | 92 | 92 | 92 | 55 | 41 | 6 | 26 | 76 | 94 | 73 | 97 | 93 | 42 | 10 | 53 | 63 |
| Zoom | 0.254 | 98 | 98 | 99 | 60 | 50 | 31 | 6 | 88 | 97 | 77 | 101 | 98 | 45 | 24 | 51 | 68 |
| Snow | 0.263 | 114 | 113 | 118 | 115 | 91 | 87 | 87 | 5 | 74 | 97 | 84 | 117 | 54 | 40 | 52 | 83 |
| Frost | 0.259 | 109 | 111 | 112 | 118 | 87 | 99 | 89 | 40 | 9 | 69 | 60 | 108 | 57 | 28 | 84 | 79 |
| Fog | 0.254 | 117 | 116 | 118 | 136 | 115 | 100 | 86 | 76 | 79 | 2 | 58 | 57 | 73 | 88 | 72 | 86 |
| Bright | 0.235 | 108 | 111 | 110 | 142 | 113 | 112 | 104 | 86 | 88 | 75 | 7 | 100 | 72 | 60 | 55 | 89 |
| Contr | 0.282 | 96 | 98 | 100 | 131 | 103 | 97 | 87 | 82 | 76 | 50 | 72 | 2 | 66 | 53 | 50 | 78 |
| Elastic | 0.200 | 124 | 122 | 125 | 127 | 92 | 91 | 72 | 92 | 110 | 98 | 104 | 118 | 19 | 22 | 61 | 92 |
| Pixel | 0.251 | 99 | 98 | 102 | 129 | 87 | 96 | 93 | 93 | 95 | 90 | 86 | 108 | 53 | 5 | 42 | 85 |
| Jpeg | 0.256 | 86 | 86 | 88 | 143 | 110 | 118 | 113 | 98 | 102 | 120 | 101 | 129 | 66 | 32 | 11 | 94 |

Table 8: CE scores computed with models trained with a data augmentation with one corruption of ImageNet-NOC. Each line refers to one model trained with one corruption of ImageNet-NOC and each column refers to one corruption of ImageNet-NOC.

| | Clean | Quant | Blur | Vert-Arti | Rain | Border | Shear | Bright | Hue | mCE |
|---|---|---|---|---|---|---|---|---|---|---|
| standard | 0.247 | 87 | 122 | 77 | 71 | 67 | 79 | 117 | 88 | 89 |
| Quant | 0.237 | 9 | 125 | 77 | 90 | 70 | 90 | 124 | 80 | 83 |
| Blur | 0.251 | 79 | 7 | 61 | 72 | 82 | 90 | 132 | 106 | 79 |
| Vert-Arti | 0.242 | 79 | 133 | 4 | 63 | 80 | 84 | 119 | 104 | 83 |
| Rain | 0.231 | 86 | 131 | 78 | 2 | 73 | 82 | 115 | 97 | 83 |
| Border | 0.241 | 89 | 123 | 71 | 73 | 8 | 74 | 118 | 89 | 81 |
| Shear | 0.234 | 129 | 127 | 65 | 85 | 75 | 6 | 133 | 102 | 90 |
| Bright | 0.236 | 75 | 125 | 78 | 77 | 73 | 92 | 21 | 89 | 79 |
| Hue | 0.261 | 83 | 139 | 91 | 96 | 74 | 101 | 116 | 1 | 88 |

