# OpenReview forum: "Increasing the Coverage and Balance of Robustness Benchmarks by Using Non-Overlapping Corruptions"
_ICLR.cc/2021/Conference — Reject_

### Official Review · AnonReviewer1 · 2020-10-26
**ImageNet-NOC**

**Rating:** 4
**Confidence:** 5

**Review:**

Summary:
This paper points out that ImageNet-C, the de facto standard for measuring robustness to natural corruptions for ImageNet classification models, contains correlated corruptions, so a mean robustness score over the ImageNet-C corruptions is biased in favor of certain classes of corruptions. It proposes two metrics: robustness score and overlapping score, and uses them to specify desirable characteristics of a robustness benchmark: coverage and balance. It specifies an algorithm for selecting a set of corruptions that optimize for these characteristics, and introduces a new benchmark (ImageNet-NOC) built using this algorithm and evaluates some models that performed well on ImageNet-C on ImageNet-NOC to see if they still perform well.

Positives:
* I think the authors are correct that some corruptions in ImageNet-C are correlated and therefore, using a simple mean over corruptions is going to bias towards models that improve on the correlated corruptions.
* Results like Section 5.4 seem valuable, since they show that existing ImageNet-C results may not generalize to robustness to a broader set of corruptions.

Concerns:
* The ImageNet-C paper specifically says "networks should not be trained on these images," for good reason, since training on them makes it too easy to optimize for the metric, but miss the overall goal. This paper specifically trains on it. Maybe it's ok for this particular case of analyzing the benchmark itself, but that should be called out very clearly. The entire benchmark becomes useless if people start training on it.
* There is an assumption made throughout the paper that using a corruption as data augmentation during training imparts robustness to that corruption on the trained model. This needs to be established. I have seen empirically that it isn't always the case.
* I worry that Algorithm 1 is itself not robust to changes in architecture or training methodology. You do present results showing that the corruptions selected using ResNet18 also have low overlap when measured on DenseNet, but I would have also liked to see whether running Algorithm 1 with the same set of candidate corruptions, but using DenseNet, would result in the same set of corruptions being selected.
* The claims in section 5.2 seem entirely misleading. The results you show are purely a result of the initial set of candidate corruptions you chose, and you provided no methodology for how to perform that step. If your initial set had been exactly ImageNet-C or even a subset of ImageNet-C then the results would have looked entirely different. Similarly, I could introduce a new corruption that isn't in your candidate set and suddenly ImageNet-NOC would look like it has low coverage. Given that the space of possible corruptions is unbounded, this notion of coverage doesn't seem at all valuable to me.
* On the whole, I think this paper would be a lot stronger if the proposal was just a set of weights for each ImageNet-C corruption, such that the top-line robustness metric used would be a weighted average that is less biased towards correlated corruptions.

Minor details:
* It's kind of weird to have section 2.3 placed where it is. It would make more sense at the beginning of section 4.
* In section 3.1, the statement that R_c^m \in [0, 1] isn't strictly true. It's possible that A_c > A_clean.
* It's worth mentioning explicitly that the robustness score, as you've defined it, is only valuable when presented along with A_clean. If A_c == A_clean because both are at chance, this has optimal robustness but isn't interesting.
* R^standard is used in Eq 1 but it isn't defined until a few paragraphs later.
* Algorithm 1 is very verbose and complicated-sounding. Steps 2-6 could just have been written as: "Pick the largest subset with overlap score under the threshold and break ties by selecting the lowest overlap score."
* There are a number of misspellings and grammatical issues throughout.

Reasons for score:
The critique of ImageNet-C is valid, but I don't think the proposed solution really addresses it. A good solution needs to be totally agnostic to architecture and training algorithm, which this method is not. There also needs to be a lot less dependence on the manually selected list of initial corruption candidates. I think ImageNet-C made a conscious choice not to involve model training when building their benchmark dataset and they had good reasons for doing so. Those reasons (laid out in the "Concerns" above) are why I don't think this paper is worthy of acceptance.

---

> ### Author Response · Authors · 2020-11-24
> **Discussions About the Benchmark Coverage**
>
> *“The claims in section 5.2 seem entirely misleading. The results you show are purely a result of the initial set of candidate corruptions you chose, and you provided no methodology for how to perform that step. If your initial set had been exactly ImageNet-C or even a subset of ImageNet-C then the results would have looked entirely different.”*
> >We agree that our results depend on the initial set of candidate corruptions we choose. The larger the initial set, the more combinations of non-overlapping corruptions can be found by the Algorithm 1, the more it increases the chances to build a large benchmark for a given overlapping_threshold. Then, if we added corruptions in the initial set of candidates displayed in Figure 2, it would make the Algorithm 1 build benchmarks that are larger i.e. that have a larger coverage. We have added information about the importance of the selection of the initial set of corruptions in Section 4.
> We agree that if we had chosen a different initial set of corruptions, the results in Section 5.2 would have been different. But our point here is to show that with the set of candidates displayed in Figure 2, the Algorithm 1 is already able to find benchmarks such as ImageNet-NOC, that are balanced and cover a large range of corruptions.
>
> *“Similarly, I could introduce a new corruption that isn't in your candidate set and suddenly ImageNet-NOC would look like it has low coverage.”*
> >We agree that you could select a corruption that is not covered by ImageNet-NOC. But we argue that it is hard to select a corruption that is not covered by ImageNet-NOC but covered by ImageNet-C. To support this statement, we propose to consider all the overlapping scores computed with one ImageNet-NOC corruption and one ImageNet-C corruption (Figure 3 has been updated in the manuscript to display these scores). It appears that all the ImageNet-C corruptions overlap with at least one ImageNet-NOC corruption. Then, an introduced corruption that overlaps with the ImageNet-C corruptions is likely to also overlap with the ImageNet-NOC corruptions. In other words, a corruption that is covered by ImageNet-C is likely to be also covered by ImageNet-NOC. This mean that it would be difficult to find a corruption that is not covered by ImageNet-NOC but covered by ImageNet-C. On the other hand, ImageNet-NOC does contain corruptions that are not covered by ImageNet-C and that is why we think that ImageNet-NOC has a larger coverage than ImageNet-C.
>
>
> *“Given that the space of possible corruptions is unbounded, this notion of coverage doesn't seem at all valuable to me.”*
> >To make the value of the notion of coverage more apparent, we propose to proceed to an additional experiment. We have computed the CE score of the m_IC model and the standard model to the *obstruction* and the *gray_scale* corruptions (described in Figure 5 and 6). The m_IC and standard models respectively have an *obstruction* CE score of 135 and 112 and a *gray_scale* CE score of 86 and 76. In other words, being robust to ImageNet-C does not imply being robust to these corruptions. This result is not surprising because we show in Section 5.2 that being robust to ImageNet-C does not imply being robust to the *hue* and *border* corruptions, and we see in Appendix A that *hue* and *border* respectively overlap with *color_distortion* and *obstruction*.
> Then, knowing that a corruption (such as *hue*) is not covered by a benchmark, does not simply mean that this benchmark does not provide a robustness guarantee only on this single corruption among the unbounded space of corruption. It means that this benchmark does not provide any robustness guarantee on a subset of the space of possible corruptions. For instance in our example, it seems that ImageNet-C does not provide any guarantee on the robustness to the corruptions that distort the colors in images. Then, we think that the coverage has a practical value: a corruption that is not covered by a benchmark, implies that this benchmark gives no information about the robustness towards a subset of the unbounded space of possible corruptions.
>
>
> *“On the whole, I think this paper would be a lot stronger if the proposal was just a set of weights for each ImageNet-C corruption, such that the top-line robustness metric used would be a weighted average that is less biased towards correlated corruptions.”*
> >We disagree with this statement because it would not solve the problem that ImageNet-C does not cover some aspects of the common corruption robustness. We demonstrated in Section 5.2 that ImageNet-C does not cover the *hue* and *border* corruptions and we also show in the answer to the last concern, that the *gray_scale* and *obstruction* corruptions are not covered by ImageNet-C either. A set of weights for each ImageNet-C corruption would not help to cover more corruptions.
>
> We thank you for the valuable feedbacks made in the minor details. We have followed your suggestions in the updated manuscript.

---

> ### Author Response · Authors · 2020-11-24
> **Discussing the Link Between Data Augmentation and Robustness**
>
> Thank you for your review and comments. We provide a point-by-point response to individual concerns below.
>
> *“The ImageNet-C paper specifically says "networks should not be trained on these images"”*
> >We have added a few lines in the Section 5.1 of the updated manuscript to indicate that a network should in general not be trained on images corrupted with the ImageNet-C corruptions. As you say, we proceed this way in our study only because we analyze the benchmark itself.
>
> *“There is an assumption made throughout the paper that using a corruption as data augmentation during training imparts robustness to that corruption on the trained model. This needs to be established. I have seen empirically that it isn't always the case.”*
> >We think that “using a corruption as data augmentation during training imparts robustness to that corruption on the trained model” is an established fact. The studies presented in Table 3 in [1], Table 1 in [2] and in Figure 4 in [3] support this statement. Besides, the diagonal cells of Table 7 and Table 8 in our paper, also show that every model is robust against the corruption used in the data augmentation process.
>
> *“I worry that Algorithm 1 is itself not robust to changes in architecture or training methodology. You do present results showing that the corruptions selected using ResNet18 also have low overlap when measured on DenseNet, but I would have also liked to see whether running Algorithm 1 with the same set of candidate corruptions, but using DenseNet, would result in the same set of corruptions being selected.”*
> >We have run the Algorithm 1 with the same settings as the ones used to get ImageNet-NOC (same set of candidates and overlapping_threshold equals to 0.1), but using the DenseNet-121 architecture instead of the ResNet18 one. The overlapping scores computed in the step (1) of the algorithm are displayed in Appendix A. The benchmark obtained after having ran the next steps of the algorithm is (*rain, hue, shear, elastic, Gaussian noise, border, brightness, vertical_artifacts*).
>
> >This benchmark contains the same number of corruptions as ImageNet-NOC, and shares six corruptions with it. The *Gaussian noise* and *elastic* corruptions replace the *quantization* and *blur* corruptions of ImageNet-NOC. Interestingly, we notice in Appendix A that *Gaussian noise* and *elastic* respectively overlap with *quantization* and *blur*. Then, the only two corruptions that have changed in the obtained benchmark are correlated in terms of robustness to the ones they replaced. In summary, the set of corruptions selected by Algorithm 1 with DenseNet-121 is very close to the one selected with ResNet18. We have added information about this experiment in the Section 4 of the updated manuscript.
>
> [1] Igor Vasiljevic, Ayan Chakrabarti, and Gregory Shakhnarovich. Examining the impact of blur on recognition by convolutional networks .arXivpreprintarXiv:1611.05760, 2016
>
> [2] Zhou, Y., Song, S., Cheung, N.: On classification of distorted images with deep convolutional neural networks. In IEEE International Conference on Acoustics, Speech and Signal Processing, 2017
>
> [3] Micha Koziarski and Boguslaw Cyganek. Image recognition with deep neural networks in presence of noise dealing with and taking advantage of distortions. Integrated Computer-Aided Engineering, 2017

---

### Official Review · AnonReviewer2 · 2020-10-28
**Official Blind Review2**

**Rating:** 5
**Confidence:** 3

**Review:**

##################################################################

Summary:

This paper proposes an algorithm to build a benchmark of Non-Overlapping Corruptions. The proposed ImageNet-NOC dataset is balanced and also covers a wider range of corruptions.

##################################################################

Pros:

(1) This paper first provides detailed analysis of dataset coverage and balance.
(2) This paper develops an algorithm to construct a robust benchmark that is balanced and non-overlapping, which means a lot for future work.
(3) This paper is well-organized relatively.
(4) The experimental results are relatively consistent with the mentioned features.

##################################################################

Cons:

(1) The coverage (Sec 5.3) is a little bit confusing. It reads “We observe that the mINOC model is 【significantly】 more robust than the standard model to all the ImageNet-C corruptions … … ImageNet-NOC covers all the corruptions of ImageNet-C”The reviewer noticed that gaps of some corruptions are not sufficiently significant, for example, Elastic (67 vs 63), Pixel (78 vs 71) and Jpeg (56 vs 53). Although marginal gaps are observed, can we conclude that ImageNet-NOC already covers that of ImageNet-C? The reviewer suggests adding the following experiments: Reporting both the results of m_IC evaluated on ImageNet-C & ImageNet-NOC, and that of m_INOC evaluated on ImageNet-C & ImageNet-NOC. Comparing “m_IC on ImageNet-C” and “m_INOC on ImageNet-C”. If these two results are similar, then we can conclude that ImageNet-NOC covers ImageNet-C.

(2) The experiments in Sec 5.4 show that SIN+IN is more robust than ANT on ImageNet-NOC. But on ImageNet-C, ANT is more robust. Is this phenomenon caused by imbalanced corruptions of ImageNet-C? Could the authors explain this? (Maybe based on the proposed overlapping scores between corruptions.)


##################################################################
[Edit]
After reading the authors' response and other reviewers' comments, I have lowered the scored to 5. The reasons are bellow.

This paper provides good analysis of the coverage and balance of robustness benchmarks. Based on the analysis, the paper presents a method to construct a benchmark of Non-Overlapping Corruptions. I think such analysis is valuable and interesting, despite some concerns about the coverage comparisons (which are only partially addressed by the authors' response).

I agree with Reviewer#1 that the comparisons between ImageNet-C and ImageNet-NOC are not solid. And that "the paper should have done an equivalent evaluation, comparing ImageNet-C as a whole vs. ImageNet-NOC as a whole." Since the core idea of this paper is to propose a better robustness benchmark, solid comparisons between ImageNet-C and ImageNet-NOC are critical.

I also agree with Reviewer #3 and Reviewer #4 that, the improvements provided by ImageNet-NOC does not make a significant difference.

The reviewer appreciates the analysis of the coverage and balance of the robustness benchmarks, which in my opinion, is valuable. However, since this paper focuses on proposing a better benchmark, solid evaluations between these two benchmarks are critical.

---

> ### Author Response · Authors · 2020-11-24
> **Providing Additional Experiments and Discussions about the ImageNet-NOC Coverage**
>
> We thank the reviewer for its feedbacks. We explain below how we have answered its legitimate questions.
>
> (1) *“The coverage (Sec 5.3) is a little bit confusing. It reads “We observe that the mINOC model is 【significantly】 more robust than the standard model to all the ImageNet-C corruptions … … ImageNet-NOC covers all the corruptions of ImageNet-C”The reviewer noticed that gaps of some corruptions are not sufficiently significant, for example, Elastic (67 vs 63), Pixel (78 vs 71) and Jpeg (56 vs 53). Although marginal gaps are observed, can we conclude that ImageNet-NOC already covers that of ImageNet-C? The reviewer suggests adding the following experiments: Reporting both the results of m_IC evaluated on ImageNet-C & ImageNet-NOC, and that of m_INOC evaluated on ImageNet-C & ImageNet-NOC. Comparing “m_IC on ImageNet-C” and “m_INOC on ImageNet-C”. If these two results are similar, then we can conclude that ImageNet-NOC covers ImageNet-C.”*
> >We agree that the term *“significantly”* in the noted sentence, is indeed not appropriate to describe our results and we removed it from the updated manuscript. Indeed there are some gaps between the CE scores that are low. But we notice that this occur when the standard model already has a low CE score. These gaps may be lower than the other gaps because it is harder to make an observable difference when the reference already has a relatively low CE score.
>
> >To make more apparent that the ImageNet-C corruptions are covered by ImageNet-NOC, we have carried out another experiment. We have computed all the overlapping scores between one ImageNet-C corruption and one ImageNet-NOC corruption (the Figure 3 has been updated to show these scores). We observe that all the ImageNet-C corruptions, overlap with at least one ImageNet-NOC corruption. For instance, it appears that the *jpeg_compression* corruption overlaps with the *quantization* corruption. On the other hand, the *border* and *hue* corruptions are not covered by any of the ImageNet-C corruption. These results are consistent with the ones obtained in Table 1 and support the idea that a model robust to ImageNet-NOC should be fairly robust to ImageNet-C. This point is discussed in the Section 5.2 of the updated version of the manuscript.
>
> >When it comes to the proposed experiment, it is true that m_INOC is not as robust as m_IC to ImageNet-C: fitting completely ImageNet-NOC does not imply fitting completely ImageNet-C. But the study of the overlapping scores and the results in Table 1, suggest that increasing the robustness to ImageNet-NOC should in practice increase the robustness to ImageNet-C.
>
>
> (2) *“The experiments in Sec 5.4 show that SIN+IN is more robust than ANT on ImageNet-NOC. But on ImageNet-C, ANT is more robust. Is this phenomenon caused by imbalanced corruptions of ImageNet-C? Could the authors explain this? (Maybe based on the proposed overlapping scores between corruptions.)”*
> >We think that this phenomenon is indeed due to the imbalanced corruptions of ImageNet-C. In Table 2.a, the average mCE score is 76, but all the models trained with a data augmentation on noise has a mCE of 71. Then, it appears that ImageNet-C gives more importance to the robustness to noises than to other of its corruptions. Interestingly, the experiments carried out in [1] (Table 9) show that Adversarial Noise Training makes models particularly robust to noises. We believe this explain why ANT is very robust to ImageNet-C, but not so robust to ImageNet-NOC which is more balanced. We haved added some discussions about this point in the manuscript.
>
> [1] Evgenia Rusak, Lukas Schott, Roland S. Zimmermann, Julian Bitterwolf, Oliver Bringmann, Matthias Bethge, and Wieland Brendel. A simple way to make neural networks robust against diverse image corruptions. ECCV, 2020.

---

### Official Review · AnonReviewer4 · 2020-10-28
**A principled approach towards robustness evaluation with some shortcomings**

**Rating:** 6
**Confidence:** 4

**Review:**

## Paper summary
The paper considers the problem of measuring the robustness of image classification models to common image perturbations. Datasets of corrupted images, such as ImageNet-C, have been created for this purpose. However, these datasets have been created from an ad-hoc, heuristic selection of perturbations. The present paper proposes a systematic approach to select types of perturbations in a way that spans a large variety of perturbations and assigns similar importance to each perturbation. Similarity of perturbations is measured based on how much training on one perturbations confers robustness against another perturbation. The paper provides an algorithm for selecting perturbations to include, and uses the algorithm to create a variant of ImageNet-C with improved coverage and balance.


## Arguments for acceptance
1. Robustness of computer vision models to image perturbations is a topic of great interest, but methods to measure robustness are either highly artificial (adversarial robustness) or ad hoc heuristics. The present paper takes an important step in the direction of making evaluation of robustness to non-adversarial corruptions more principled.
2. The paper is written clearly.
3. The overlapping score is based on a practically relevant quantity, namely the performance of a network on the corrupted data.
4. A simple algorithm is provided for selecting perturbations for new datasets.
5. A new alternative to ImageNet-C is created with the proposed algorithm. The new dataset has improved coverage and balance properties. Some evidence is provided that these improvements can affect the results of robustness evaluations.

## Arguments against acceptance
6. The computational cost of the method is high, and not stated or discussed. As far as I can tell, at least one neural network needs to be trained for each candidate dataset. Although the authors state that results transfer across architectures, such that a small architecture can be used, it is known that robustness properties depend significantly on model size and training procedure (e.g. see https://arxiv.org/abs/2007.08558). This should be addressed further.
7. This method only really works for synthetic corruptions for which many new examples can be generated. Otherwise, it may be difficult to obtain enough examples to train the network used for computing the overlapping score.
8. It is not discussed whether Algorithm 1 is guaranteed to provide the optimal combination of datasets (in terms of overlap and balance).
9. It is not entirely clear if the improvements provided by ImageNet-NOC make a significant difference in practice. Table 3 starts to address this question, but it would be useful to compare ImageNet-C and ImageNet-NOC mCE across a wider range of models (e.g. pretrained models available online). What is the rank correlation between ImageNet-C and ImageNet-NOC mCE?

## Conclusion and suggestions
This is a borderline submission. Because of the principled approach to an important and topical problem, I tend towards accepting it.

Suggestions for improvement:
10. Discuss the computational cost of the method.
11. Discuss the optimality of Algorithm 1.
12. Compare ImageNet-C and ImageNet-NOC on a wider range of models.

---

> ### Author Response · Authors · 2020-11-24
> **Extending the Discussions about the Algorithm and Adding Comparisons between the Benchmarks**
>
> We thank the reviewer for the detailed review and valuable suggestions. Below is described the way we have addressed the reviewer’s concerns.
>
> 6)
> *“The computational cost of the method is high, and not stated or discussed. As far as I can tell, at least one neural network needs to be trained for each candidate dataset.”*
> >Indeed, the computational cost of the method is high because it requires to train one model for each corruption in the initial set of corruptions. Training one ResNet18 for every corruption displayed in Figure 2, took one week with a single GPU Nvidia Tesla V100. We think that the computational cost of the method could be significantly reduced by fine-tuning models during a few epochs instead of training them from scratch. Unfortunately, we have not succeeded in verifying in time whether the results obtained with fine-tuning are significantly different from the ones obtained with complete trainings. We have added this information in the Section 4 of the updated manuscript.
>
> *“Although the authors state that results transfer across architectures, such that a small architecture can be used, it is known that robustness properties depend significantly on model size and training procedure (e.g. see https://arxiv.org/abs/2007.08558). This should be addressed further.”*
> >We agree that the robustness of models is largely dependent on the model size and the training procedure. We also agree that if we had used a very small architecture, too small to fit the training data for instance, we would have obtained very different overlapping scores. However, we have computed the overlapping scores between all the ImageNet-NOC and ImageNet-C corruptions with the ResNet18, DenseNet-121 and Wide-ResNet-50-2 architectures. The results obtained are displayed in Appendix D of the updated manuscript. It appears that even if these three models do not have the same size or structure, the obtained overlapping arrays are very similar. We think the reason we obtain these results is that the overlapping metric relates to the robustness gain of models due to a data augmentation rather than the robustness of models itself. So, the study of the corruption overlappings, may depend less on the architecture and on the training procedure of models than the robustness metrics.
>
> 7)
> >Indeed this is a constraint of our method. In this case, one option is to use a synthetic corruption as a substitute for the corruption for which we cannot obtain many examples, and assume that this substitute overlaps with the corruption it replaces. We think it could be feasible because Hendrycks et al [1] showed that synthetic corruptions can help to deal with real-world distribution shifts. However, we are aware that this solution could be complicated and very time-consuming to implement in practice.
>
> 8)
> >For each benchmark b of n corruptions provided by Algorithm 1, it is guaranteed that b is the n-corruption benchmark with the lowest mean overlapping score as possible. In other words, each n-corruption benchmark obtained with the Algorithm 1, is the optimal benchmark in terms of overlapping among all the possible n-corruption benchmarks. When it comes to the balance, we have not established a direct link between the optimality in terms of overlapping and the optimality in terms of balance. So, we cannot guarantee that Algorithm 1 provides optimal benchmarks in terms of balance. However, as argued in Section 3.3, we think that minimizing the overlappings in benchmarks should maximize their balance, and we show empirically in Section 5.3 that a benchmark obtained with Algorithm1, is clearly more balanced than one that is not optimal in terms of overlappings. We have added explanations in Section 4 to make this point clearer.
>
> 9)
> >We have provided more results and discussions in Section 5.4 to help to compare ImageNet-NOC with ImageNet-C. In particular, we have added in Table 4 the results obtained with *AugMix* and *DeepAugment* on ImageNet-NOC and ImageNet-C. We have also compared in Table 3, the ImageNet-NOC mCE scores with the ImageNet-C ones using various pretrained torchvision models. We see in Table 4 that the robustness ranking established with ImageNet-NOC is not aligned with the one obtained with ImageNet-C. In Table 3, it appears that the VGG-19, ResNet-50, and WideResNet-50-2 are much more robust to ImageNet-NOC than the AlexNet model, but they are less robust than AlexNet to ImageNet-C. These results confirm that using ImageNet-NOC instead of ImageNet-C makes a difference in practice.
>
> [1] Dan Hendrycks, Steven Basart, Norman Mu, Saurav Kadavath, Frank Wang, Evan Dorundo, Rahul Desai, Tyler Zhu, Samyak Parajuli, Mike Guo, Dawn Song, Jacob Steinhardt, and Justin Gilmer. The many faces of robustness: A critical analysis of out-of-distribution generalization arXivpreprint arXiv:2006.16241, 2020.

---

### Official Review · AnonReviewer3 · 2020-10-28
**Modification to ImageNet-C too marginal**

**Rating:** 5
**Confidence:** 5

**Review:**

This paper proposes a new dataset for estimating robustness to distribution shift, in particular corruption robustness. They accomplish this by proposing an alternative to ImageNet-C, ImageNet-NOC, which uses different corruptions. They consider corruptions not in ImageNet-C, and they argue that their dataset is superior because they have more "balance and coverage." They select corruptions that are "decorrelated" in a specific sense.

I appreciate the analysis in this paper and provides useful comparisons to ImageNet-C. Unfortunately experimentation is not thorough. They don't show results with AugMix nor DeepAugment, so the analysis only uses SIN and ANT.

The paper is missing key qualifiers.
"In other words, the expression (1) estimates if increasing the robustness to c2 implies an increase of robustness to c1 and vice versa."
... provided that the model fits the exact corruption c2.
"Then, the higher (1) is, the more the robustnesses to c1 and c2 are correlated, i.e. the more c1 and c2 overlap."
... provided the models are robustified through exactly training on c1 or c2.
Without these qualifications, the claims are far too general.
Contrast and fog can be negatively correlated with the rest of the corruptions under some robustness interventions. For example, adversarial training really harms contrast and fog corruption robustness, though it might help other corruptions like noise (and a less expansive testbed might not catch this). This demonstrates that their robustness correlations require qualification and are less predictive that this paper suggests.

This paper is fairly similar to Is Robustness Robust? On the interaction between augmentations and corruptions (submission #704). If these papers have very different ratings, then there's a problem with this review process.

Nitpicks:

The bibtex for the citations are messed up. For example, we see "(Evgenia Rusak, 2020)" instead of "(Rusak et al., 2020)". Author ordering for papers in the bibtex is often scrambled, some are separated by commas, and some are not. In general this paper's formatting is a little unrefined.

"weight decay set to 10e-4." -> "weight decay set to $10^{-4}$."

"For instance, a benchmark that contains a motion blur corruption covers the defocus blur corruption, because the robustnesses [sic] towards these two corruptions are correlated (Vasiljevic et al., 2016)."
Vasiljevic et al. (https://arxiv.org/pdf/1611.05760v1.pdf) show the opposite. Fine-tuning on defocus blur did made models slightly worse on horizontal motion blur (Figure 2).

---

> ### Author Response · Authors · 2020-11-24
> **Adding Comparisons between the Benchmarks and Adding Qualifications to the Metric**
>
> We thank the reviewer for its valuable remarks. We explain below how we have addressed the reviewer’s concerns.
>
> *“I appreciate the analysis in this paper and provides useful comparisons to ImageNet-C. Unfortunately experimentation is not thorough. They don't show results with AugMix nor DeepAugment, so the analysis only uses SIN and ANT.”*
> >We have added in Table 4 the results obtained with *AugMix* and *DeepAugment* on ImageNet-NOC and ImageNet-C. We have also compared the mCE scores of diverse pretrained image classifiers in Table 3. The robustness ranking established with ImageNet-C in Table 4 is not aligned with the one obtained with ImageNet-NOC. In Table 3, we show that VGG-19, ResNet-50, and WideResNet-50-2 are much more robust to ImageNet-NOC than the AlexNet model, but they are less robust than AlexNet to ImageNet-C. These results confirm that using ImageNet-NOC instead of ImageNet-C to make a robustness estimation, makes a difference in practive. We have completed the discussion in Section 5.4 to better describe and understand the obtained results.
>
>
> *“The paper is missing key qualifiers. "In other words, the expression (1) estimates if increasing the robustness to c2 implies an increase of robustness to c1 and vice versa." ... provided that the model fits the exact corruption c2. "Then, the higher (1) is, the more the robustnesses to c1 and c2 are correlated, i.e. the more c1 and c2 overlap." ... provided the models are robustified through exactly training on c1 or c2. Without these qualifications, the claims are far too general. Contrast and fog can be negatively correlated with the rest of the corruptions under some robustness interventions. For example, adversarial training really harms contrast and fog corruption robustness, though it might help other corruptions like noise (and a less expansive testbed might not catch this). This demonstrates that their robustness correlations require qualification and are less predictive that this paper suggests.”*
> >We agree that our claims were too general. Then, we have added these qualifications and rewritten the whole paragraph to answer this concern.
> That being said, we would like to provide more evidence that the overlapping score is in practice a good predictor. In particular, an overlapping score that is equal to zero, predicts that the two corruptions used to compute it, are in practice not correlated or negatively correlated. Concerning the given example about adversarial training, we see in Figure 3 that the overlapping scores computed with either *fog* or *contrast* and any noise, are equal to zero. This is consistent with the fact that adversarial training can increase the robustness to noises while diminishing the robustness to fog and contrast. Likewise, the overlapping scores in Figure 3 predict that the robustnesses towards the corruptions that alter the high-frequency information of images (*noises*, *blur*, *pixelate*, *jpeg compession*) are correlated. Here again, the prediction is consistent with the conclusions made in [1], that state that the robustnesses of neural networks towards the corruptions that degrade the high-frequency information in images are correlated. In general, we have observed that the overlapping score is often useful to predict if the robustnesses towards two corruptions are correlated in practice.
>
> *“The bibtex for the citations are messed up. For example, we see "(Evgenia Rusak, 2020)" instead of "(Rusak et al., 2020)". Author ordering for papers in the bibtex is often scrambled, some are separated by commas, and some are not. In general this paper's formatting is a little unrefined.”*
> >Thank you for pointing out the errors in the citations. We have rectified these mistakes in the updated manuscript.
>
> *“weight decay set to 10e-4.”*
> >We have corrected this typo, thank you.
>
> *“For instance, a benchmark that contains a motion blur corruption covers the defocus blur corruption, because the robustnesses [sic] towards these two corruptions are correlated (Vasiljevic et al., 2016)." Vasiljevic et al. (https://arxiv.org/pdf/1611.05760v1.pdf) show the opposite. Fine-tuning on defocus blur did made models slightly worse on horizontal motion blur (Figure 2).”*"
> >We are sorry for this error. The blurs that are shown to overlap in the latest version of the paper (https://arxiv.org/pdf/1611.05760.pdf) are the defocus blur and the camera shake blur (not the motion blur). We have rectified our mistake in the manuscript. Thank you for pointing this out.
>
> [1] Dong Yin, Raphael Gontijo Lopes, Jonathon Shlens, Ekin D. Cubuk, and Justin Gilmer. A fourier perspective on model robustness in computer vision. ICML Workshop on Uncertainty and Robustness in Deep Learning, 2019.

---

### Author Response · Authors · 2020-11-25
**Revision Summary**

Here is a summary of the main concerns and how we have addressed them:

**Not enough comparisons between ImageNet-C and ImageNet-NOC.**

We have provided several additional comparisons in Section 5.4 to show that using ImageNet-NOC instead of ImageNet-C makes a difference in practice.

**Results obtained are not agnostic to architecture.**

We have carried out additional experiments in Section 4 and 5 to show that the overlapping scores and the results of Algorithm 1 do not vary much with the used architecture.

**Information about Algorithm 1 is missing.**

We have added discussions about: (1) its computational cost, (2) the influence of the initial selection of candidate corruptions on the results, (3) the optimality of the obtained benchmarks in terms of overlappings (4), the influence of the used architecture on results.

**It is not clear that ImageNet-NOC covers ImageNet-C.**

We have computed additional overlapping scores (displayed in Figure 3) and show in Section 5.2 that these scores are consistent with the coverage study conducted in the same section.

---

### Decision · Program_Chairs · 2021-01-07
**Final Decision**

**Decision:**

Reject

**Comment:**

The authors propose a new dataset, namely ImageNet-NOC, for evaluating robustness of image classifiers to corruptions. The dataset may be viewed as an alternative to ImageNet-C which uses a different set of corruptions. To derive this set of corruptions, the authors first develop a notion of similarity between two corruptions, and then propose an iterative algorithm to build a set of corruptions which, intuitively, is sufficient to cover the larger set of corruptions (i.e., enjoys *high coverage*), and assigns a similar importance to each such corruption (i.e., is *balanced*). Then, the authors argue that ImageNet-NOC is superior to ImageNet-C as it achieves a higher degree of balance and coverage.

The reviewers found this to be a borderline paper. The reviewers appreciated the introduced metric and agree that there is no point in evaluating on corruptions which are perfectly correlated. In addition, the systematic approach for generating a set of relevant corruptions is seen as a step in the right direction. The reviewers appreciated the author response and were engaged in the discussion. As it currently stands the reviewers are not convinced that the paper is ready for acceptance. To improve the manuscript the authors could extend Tables 3 and 4 with a wider range of models and investigate qualitative differences between models robust on one dataset, but not on the other. Furthermore, there should be a more detailed discussion of stability and computational properties of algorithm 1. In addition, the authors should provide strong arguments as to why is it not sufficient to add additional corruptions to ImageNet-C and compute a weighted score instead. The latter suggestion could lead to an iterative improvement of the current set of benchmarks and place more emphasis on the methodology. I suggest the authors to incorporate the reviewer's feedback and place more emphasis on the methodology around algorithm 1, rather then on introducing another dataset which is likely to be superseded as soon as we add a couple more corruptions in the mix.